# Cul3-KLHL20 E3 ubiquitin ligase plays a key role in the arms race between HIV-1 Nef and host SERINC5 restriction

Sunan Li[1,4], Rongrong Li[1,4], Iqbal Ahmad[1], Xiaomeng Liu[1], Silas F. Johnson [2], Liangliang Sun [3] & Yong-Hui Zheng [1,2✉]

HIV-1 must counteract various host restrictions to establish productive infection. SERINC5 is a potent restriction factor that blocks HIV-1 entry from virions, but its activity is counteracted by Nef. The SERINC5 and Nef activities are both initiated from the plasma membrane, where SERINC5 is packaged into virions for viral inhibition or downregulated by Nef via lysosomal degradation. However, it is still unclear how SERINC5 is localized to and how its expression is regulated on the plasma membrane. We now report that Cullin 3-KLHL20, a *trans*-Golgi network (TGN)-localized E3 ubiquitin ligase, polyubiquitinates SERINC5 at lysine 130 via K33/K48-linked ubiquitination. The K33-linked polyubiquitination determines SERINC5 expression on the plasma membrane, and the K48-linked polyubiquitination contributes to SERINC5 downregulation from the cell surface. Our study reveals an important role of K130 polyubiquitination and K33/K48-linked ubiquitin chains in HIV-1 infection by regulating SERINC5 post-Golgi trafficking and degradation.

[1] Harbin Veterinary Research Institute, CAAS-Michigan State University Joint Laboratory of Innate Immunity, State Key Laboratory of Veterinary Biotechnology, Chinese Academy of Agricultural Sciences, Harbin, China. [2] Department of Microbiology and Molecular Genetics, Michigan State University, East Lansing, MI, USA. [3] Department of Chemistry, Michigan State University, East Lansing, MI, USA. [4] These authors contributed equally: Sunan Li, Rongrong Li. ✉email: zhengyo@msu.edu

Mammalian cells have evolved complex defense mechanisms to restrict viral infection, but HIV-1 has also evolved various antagonisms to counteract different host restrictions to establish productive infection[1]. Serine incorporator 5 (SERINC5) is a potent restriction factor that was discovered to function in the intrinsic immunity that defenses against retrovirus infection[2,3]. SERINC5 is a ~45-kDa integral membrane protein that has ten transmembrane helices organized into two subdomains[4]. It is incorporated into budding virus particles from the cell surface of infected cells and subsequently disrupts the viral envelope (Env) glycoprotein trimmers in their open conformation, which prevents the fusion pore formation between virion and target cell, and thereby blocks virus entry[5–8]. SERINC5 antiviral activity is conserved across different species, and its important role in defending against retrovirus infection in vivo has been demonstrated in humans, monkeys, and mice[9–11].

To evade this powerful host restriction, retroviruses antagonize SERINC5 by expressing three structurally unrelated proteins: negative factor (Nef) from human immunodeficiency virus type 1 (HIV-1), glycoGag from murine leukemia virus (MLV), and S2 from equine infectious anemia virus (EIAV)[2,3,12,13]. Nef, glyco-Gag, and S2 downregulate SERINC5 from the cell surface by directing SERINC5 to endosomes/lysosomes for degradation, which prevents SERINC5 from incorporation into virions[14–16]. In particular, Nef interacts with SERINC5 via its largest intracellular loop 4 (ICL4)[17]. In addition, Nef directs Cyclin K/Cyclin-dependent kinase 13 (CDK13) complex to phosphorylate a serine residue (S360) at SERINC5 in ICL4, which bridges SERINC5 with the adapter protein 2 (AP2) complex via Nef for endocytosis and degradation[18].

SERINC5 is expressed on the plasma membrane, where SERINC5 antiviral activity and Nef counteractive activity are initiated. When SERINC5 is dissociated from the plasma membrane by removal of its 10th transmembrane domain, its antiviral activity is lost[19]. In addition, when Nef is dissociated from the plasma membrane by removal of its myristoylation site, its downregulation of SERINC5 is also blocked[16]. However, it is still unclear how SERINC5 trafficking is controlled at the post-Golgi level and how its expression is regulated on the plasma membrane.

Ubiquitination is catalyzed by three types of enzymes, including E1 ubiquitin-activating enzymes, E2 ubiquitin-conjugating enzymes, and E3 ubiquitin ligases[20]. Polyubiquitin chains are created when seven lysine residues K6, K11, K27, K29, K33, K48, and K63, or the N-terminal methionine (M1) of the 76-amino-acid (aa) ubiquitin itself are ubiquitinated by E1, E2, and E3. The substrate specificity of ubiquitination is determined by the large number of E3s. RING (Really Interesting New Gene) E3s are the most abundant ubiquitin ligases, in which the multi-subunit Cullin-RING ubiquitin ligases (CRLs) comprise the largest family[21]. CRLs are assembled from seven Cullin (Cul) proteins (Cul1, Cul2, Cul3, Cul4A, Cul4B, Cul5, and Cul7) that serve as a scaffold that engages substrates with E2s. In CRL3 E3 ligases, Cul3 recruits E2s via RING-box protein 1 (Rbx1), and substrates by Bric-a-brac/Tramtrack/Broad (BTB) proteins[22]. Among these BTB proteins, Kelch-repeat domain subfamily is the most prevalent in metazoans, in which the trans-Golgi network (TGN)-localized, Kelch-like protein 20 (KLHL20), functions as an important Cul3 substrate adapter[23].

Here, we show that CRL3$^{KLHL20}$ is a critical SERINC5 E3 ubiquitin ligase that is required for SERINC5 plasma membrane localization, which in turn determines its virion incorporation and downregulation by Nef. Thus, CRL3$^{KLHL20}$ plays a critical role in SERINC5 intracellular trafficking, antiviral activity, and expression during HIV-1 infection.

## Results

**Identification of Cul3-KLHL20 as a SERINC5 ubiquitin E3 ligase via mass spectrometry.** To understand how SERINC5 is polyubiquitinated, we interrogated the presence of E3 ubiquitin ligases in SERINC5 protein complexes by mass spectrometry. FLAG-tagged SERINC5 was purified from HEK293T cells by an anti-FLAG affinity column and analyzed by liquid chromatography tandem mass spectrometry (LC-MS/MS), as we reported[18]. Four independent experiments were conducted from which a total of 25 ubiquitin E3 ligase-associated proteins were identified, which consisted of Cul1, Cul3, and Cul4B (Fig. 1A). Because Cul4B is expressed in the nucleus and responsible for cell cycle regulation, only Cul1 and Cul3 were selected for further study.

We reported that ectopic ubiquitin (Ub) decreases SERINC5 expression at steady state[14–16]. Accordingly, we expressed SERINC5 with Ub in HEK293T cells in the presence of Cul1 and/or its adapter protein S-phase kinase-associated protein 1 (Skp1), or Cul3 and/or its adapter protein KLHL20. The SERINC5 expression was then analyzed by western blotting (WB) (Fig. 1B). A SERINC5 lysine-free mutant with all 19 lysine residues mutated to arginine (1-19 K/R), was included as a control. Ub decreased SERINC5 expression as expected (Fig. 1B, lanes 1, 2, 6, 7). Although this decrease was not affected by single expression of Cul1, Skp1, Cul3, or KLHL-20, and co-expression of Cul1 with Skp1 (lanes 3–5, 7–8), it was notably enhanced by co-expression of Cul3 with KLHL20 (lane 10). Neither Ub nor Cul3/KLHL20 decreased the 1-19 K/R mutant expression (lanes 11–15). In addition, KLHL20 decreased the amount of detectable Ub (lanes 8, 10, 13, 15), indicating that Cul3-KLHL20 deceases the cellular Ub pool. These results suggested that SERINC5 could be targeted by CRL3$^{KLHL20}$ E3 ligase.

Next, we interrogated the mechanism of how SERINC5 interacts with Cul3-KLHL20. We tried to knock out *Cul3* and *KLHL20* in HEK293T cells after expressing Cas9 with their specific short guide (sg) RNAs. Although these sgRNAs effectively knocked out *Cul3* or *KLHL20* (Fig. 1C), we could not obtain stably knockout clones due to their necessity to the cell survival. Thus, we used the same sets of Cas9 and sgRNAs to transiently knock down these two genes and studied their interaction with SERINC5.

To confirm that SERINC5 interacts with Cul3-KLHL20, they were expressed in HEK293T cells and SERINC5 was subjected to immunoprecipitation (IP) (Fig. 1D). SERINC5 pulled down KLHL20 and Cul3, but not Green Fluorescent Protein (GFP) that served as a negative control (Fig. 1D, lanes 8, 9, 11), indicating that SERINC5 interacts with Cul3-KLHL20. To test whether these interactions were dependent on each other, *KLHL20* and *Cul3* were knocked down by CRISPR/Cas9. The SERINC5-KLHL20 interaction was not affected by *Cul3*-knockdonw (KD), but the SERINC5-Cul3 interaction was disrupted by *KLHL2*-KD (lanes 10, 12). These results demonstrate that SERINC5 interacts with Cul3-KLHL20 via KLHL20.

We then tested how SERINC5 interacts with KLHL20. KLHL20 has 609 aa that comprise three domains including Kelch-repeat (1–316 aa), BTB and C-terminal Kelch (BACK), and Bric-a-brac/tramtrack/broad complex (BTB) (301–609 aa). In addition, six residues V109, I111, D113, C146, L148, and L150 are required for its binding to Cul3[23]. We generated two KLHL20 mutants, namely ΔK by deleting the Kelch-repeat domain, and 5A by replacing V110, R112, I114, L147, and Q149 with alanine, and tested their interaction with SERINC5 by IP (Fig. 1E). Both the wild-type (WT) KLHL20 and its 5A mutant interacted with SERINC5 (Fig. 1E, IP, lanes 2, 4), whereas the ΔK mutant did not (IP, lane 3). These results demonstrate that SERINC5 binds to KLHL20 via the Kelch-repeat domain.

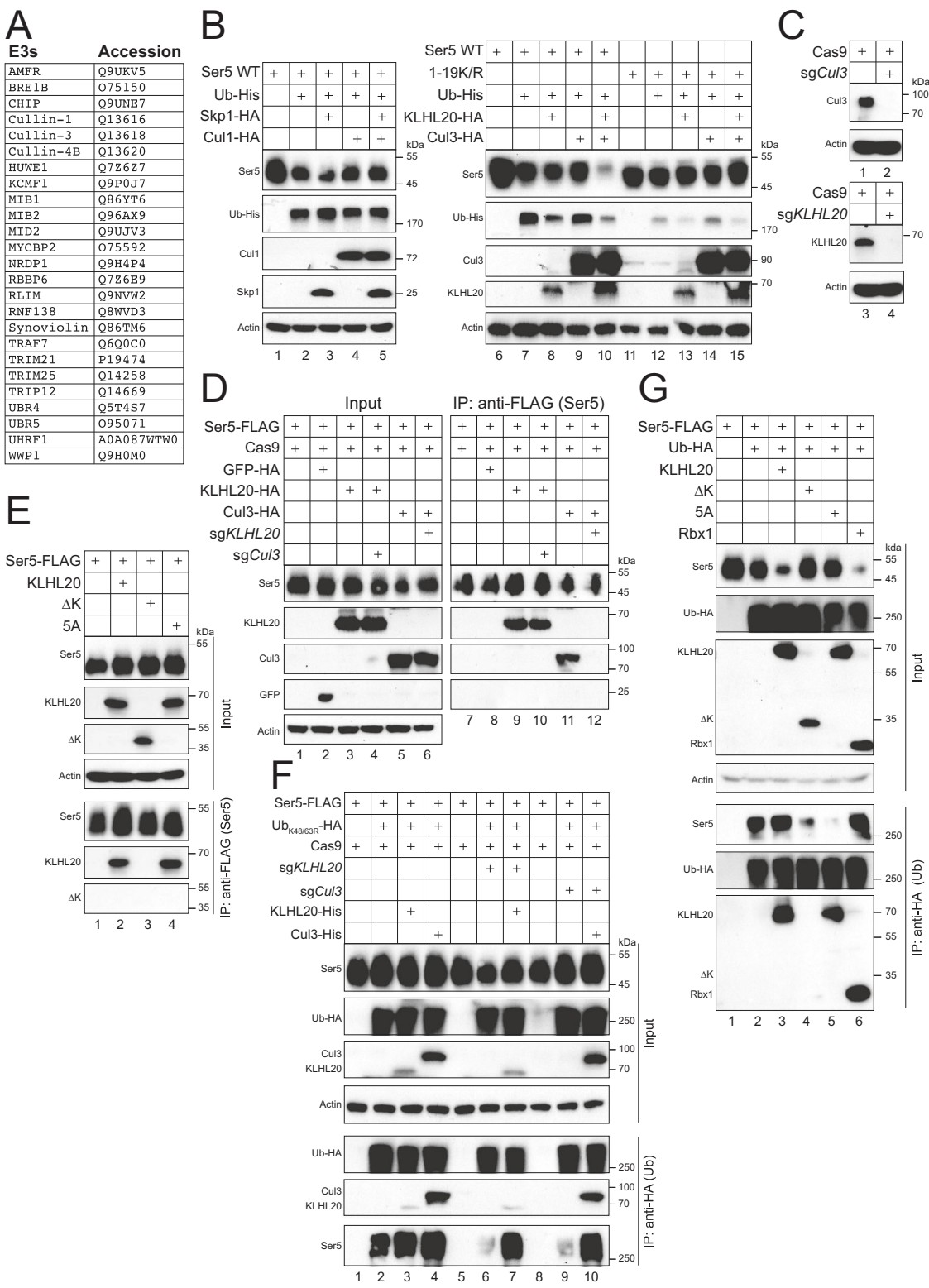

Next, we investigate the role of Cul3-KLHL20 in SERINC5 polyubiquitination. SERINC5 was expressed with Ub in HEK293T cells in the presence of KLHL20/Cul3 expression or their KDs by CRISPR/Cas9. Ectopic Ub was subjected to immunoprecipitation and levels of SERINC5 polyubiquitination were analyzed by WB (Fig. 1F). To avoid SERINC5 degradation by Ub, a Ub mutant that had its K48 and K63 mutated to arginine ($Ub_{K48/63R}$) was used. Polyubiquitinated SERINC5 proteins were detected at higher molecular mass over 100 kDa, that were

increased by KLHL20 and/or Cul3 (Fig. 1F, IP, lanes 2–4). In addition, both *KLHL20-* or *Cul3*-KD notably reduced the SERINC5 polyubiquitination (IP, lanes 2, 6, 9). Importantly, this reduction by KDs was rescued by ectopic KLHL20 or Cul3 expression (IP, lanes 7, 10).

To further explore this mechanism, SERINC5 polyubiquitination was re-analyzed in the presence of ectopic Ub, different KLHL20 proteins, and the Cul3 E2-recruiting protein Rbx1 (Fig. 1G). When SERINC5 expression in cells was compared, its

**Fig. 1 Identification of Cul3-KLHL20 as a SERINC5 ubiquitin E3 ligase via mass spectrometry. A** A list of E3 Ub ligases that were identified from the SERINC5 protein complex by mass spectrometry. **B** Ser5 was expressed with Ub in the presence of Skp1and/or Cul1 in HEK293T cells. Similarly, Ser5 and its 1-19 K/R mutant were expressed with Ub in the presence of KLHL20 and/or Cul3. Expression of these proteins was analyzed by WB using anti-epitope tag antibodies. **C** Cas9 was expressed with *Cul3-* or *KLHL20*-specific sgRNAs in HEK293T cells. Expression of Cul3 and KLHL20 was determined by WB using their specific antibodies. **D** Ser5 was expressed with Cas9, GFP, KLHL20, Cul3, and/or *KLHL20-* or *Cul3*-sgRNAs in HEK293T cells. Ser5 was immunoprecipitated by anti-FLAG, and levels of GFP, KLHL20, and Cul3 in cell lysate (Input) and IP samples were compared by WB. **E** Ser5 was expressed with KLHL20 and its ΔK and 5A mutants in HEK293T cells. Ser5 was immunoprecipitated and levels of these different KLHL20 proteins in cell lysate and IP samples were compared by WB. **F** Ser5 was expressed with Ub mutant $Ub_{K48/63R}$, KLHL20, Cul3, and/or Cas9 plus *KLHL20-* or *Cul3*-sgRNAs in HEK293T cells. Ectopic Ub was immunoprecipitated by anti-HA and levels of polyubiquitinated Ser5 were compared by WB. **G** Ser5 was expressed with Ub, KLHL20 or its mutants ΔK and 5A, and Rbx1 in HEK293T cells. Ectopic Ub was immunoprecipitated by anti-HA and levels of polyubiquitinated Ser5 were compared by WB. Ser5, serine incorporator 5; Cul, cullin; Ub, ubiquitin; Skp1, S-phase kinase-associated protein 1; KLHL20, Kelch-like protein 20; Cas9, CRISPR associated protein 9; sg, short guide RNA; GFP, green fluorescent protein; Rbx1, RING-box protein 1; ΔK, KLHL20 mutant with Kelch-repeat domain deleted; 5A, KLHL20 mutant with V110, R112, I114, L147, and Q149 mutated to alanine; HA, hemagglutinin-tag; His, polyhistidine-tag; FLAG, FLAG-tag. All experiments were repeated at least twice, and similar results were obtained. Source data are provided as a Source Data file.

decrease by Ub was enhanced by KLHL20 and Rbx1 (Fig. 1G, Input, lanes 3, 6), but not by KLHL20 mutants ΔK and 5A (Input, lanes 4, 5). Consistently, when SERINC5 polyubiquitination was compared, it was increased by KLHL20 and Rbx1 (IP, lanes 3, 6), but was strongly decreased by these two KLHL20 mutants (IP, lanes 4, 5). These results demonstrate that like KLHL20, Rbx1 is also required for SERINC5 polyubiquitination. In addition, SERINC5 polyubiquitination depends on the Cul3-KLHL20 interaction and the Kelch-repeat domain. Furthermore, it was noticeable that the ΔK mutant was not detected from the pulldown sample (IP, lane 4), which further confirms that SERINC5 interacts with KLHL20 via the Kelch-repeat domain. Altogether, these results demonstrate that CRL3[KLHL20] is responsible for SERINC5 polyubiquitination.

**Identification of lysine 130 (K130) as a critical polyubiquitination site on SERINC5.** The 423 aa of human SERINC5 comprise 5 extracellular loops (ECLs), 10 transmembrane domains (TMDs), and 4 intracellular loops (ICLs) (Fig. 2A). They also include 19 lysine residues (K1 to K19) that are spread throughout these regions. To test if these lysine residues are targeted for polyubiquitination, we generated SERINC5 mutants in which these lysine residues were replaced with arginine and determined how levels of SERINC5 polyubiquitination are affected.

Initially, we tested how ubiquitin affects expression and polyubiquitination of mutants 1-7 K/R, 1-11 K/R, 1-14 K/R, 15-19 K/R, and 1-19 K/R, which have K1-K7, K1-K11, K1-K14, K15-K19, or K1-K19 replaced with arginine. These SERINC5 lysine mutants were expressed with ectopic Ub and immunoprecipitated by anti-FLAG that targets SERINC5, and their polyubiquitination was detected by anti-HA that detects ectopic Ub (Fig. 2B). Ub reduced expression of SERINC5 WT, 1-7 K/R, and 15-19 K/R in cells, and consistently, high levels of polyubiquitinated SERINC5 products were detected from these proteins (Fig. 2B, lanes 1–4, 9–10). On the contrary, Ub did not affect expression of 1-11 K/R, 1-14 K/R, and 1-19 K/R in cells, and consistently, their polyubiquitinated products were detected at much reduced levels (lanes 5–8, 11–12). These results suggest that there is a specific SERINC5 lysine residue(s) responsible for its polyubiquitination.

To further narrow down the specific polyubiquitination site, we mutated K8 and K9 at position 130 or 179 to arginine, by generating mutants K130R and K179R. When these two mutants were expressed with Ub, expression of K179R (lanes 17–18), but not K130R (lanes 15–16), was decreased in cells. In addition, K179R had a similar level of polyubiquitination as WT SERINC5 (lanes 13–14, 17–18), whereas K130R was poorly polyubiquitinated (lanes 15–16). These results demonstrate that K130 is the critical site for SERINC5 polyubiquitination.

To understand how K130 is polyubiquitinated, we generated a SERINC5 mutant, designated 130 K, in which all the lysine residues, except K130, were replaced with arginine. In addition, we generated seven Ub mutants, $Ub_{K6R}$, $Ub_{K11R}$, $Ub_{K27R}$, $Ub_{K29R}$, $Ub_{K33R}$, $Ub_{K48R}$, and $Ub_{K63R}$, where each of the seven lysine residues in Ub were individually mutated to arginine. We also generated a mutant $Ub_{KO}$ in which all seven lysine residues were mutated to arginine. When the SERINC5-130K mutant was expressed with either WT Ub ($Ub_{WT}$) or each of the respective Ub mutants, its expression was only reduced in the presence of $Ub_{WT}$ (Fig. 2C, Input, lane 2). Immunoprecipitation with anti-HA revealed that the polyubiquitination of K130 was detected with $Ub_{WT}$, but not $Ub_{KO}$, further confirming that K130 is the target for ubiquitination (IP, lanes 2, 10). Notably, although this SERINC5-130K mutant had a similar level of polyubiquitination with $Ub_{K6R}$, $Ub_{K11R}$, $Ub_{K27R}$, $Ub_{K29R}$, and $Ub_{K63R}$ as $Ub_{WT}$ (IP, lanes 3–6, 9), this level was significantly decreased with $Ub_{K33R}$ and $Ub_{K48R}$ (IP, lanes 7–8).

To confirm the important role of K33 and K48 in SERINC5 polyubiquitination, we created another seven ubiquitin mutants, $Ub_{K6}$, $Ub_{K11}$, $Ub_{K27}$, $Ub_{K29}$, $Ub_{K33}$, $Ub_{K48}$, and $Ub_{K63}$, that only express each of those seven lysine residues of Ub. When SERINC5-130K was expressed with these Ub mutants, a similar level of SERINC5 polyubiquitination was detected from $Ub_{WT}$, $Ub_{K33}$, and $Ub_{K48}$ (Fig. 2D, IP, lanes 2, 7, 8), but this level was significantly reduced from $Ub_{K6}$, $Ub_{K11}$, $Ub_{K27}$, $Ub_{K29}$, $Ub_{K33}$, $Ub_{K63}$, and $Ub_{KO}$ (IP, lanes 3–6, 9, 10). These results demonstrate that SERINC5 is preferably polyubiquitinated at K130 via K33 and K48-linked ubiquitin chains.

To confirm the role of CRL3[KLHL20] in K130 polyubiquitination, SERINC5-K130 was expressed with $Ub_{WT}$, $Ub_{K33}$, $Ub_{K48}$, and $Ub_{KO}$ in the presence or absence of *KLHL20*-KD in HEK293T cells, and K130 polyubiquitination was determined as described above. Again, we detected a similar level of SERINC5 polyubiquitination from $Ub_{WT}$, $Ub_{K33}$, and $Ub_{K48}$, but not any from $Ub_{KO}$ (Fig. 2E, IP, lanes 2, 4, 6, 8). Importantly, KLHL20-KD completely disrupted the K130 polyubiquitination (IP, lanes, 3, 5, 7). These results demonstrate that CRL3[KLHL20] is responsible for K130 polyubiquitination via K33/K48-linked ubiquitin chains.

**K130 is required for SERINC5 localization to the plasma membrane.** To understand how these lysine mutations affect SERINC5, first, we tracked SERINC5 subcellular localization by confocal microscopy. SERINC5 was fused with a GFP tag at its C-terminus and similar lysine mutations were introduced into this SERINC5-GFP fusion protein. When these proteins were expressed in HeLa cells, SERINC5-GFP WT, 1-7 K/R, and K179R were localized to the cell surface, whereas 1-14 K/R, 1-19 K/R, 1-11 K/R, and K130R were found in the cytoplasm (Fig. 3A, HeLa,

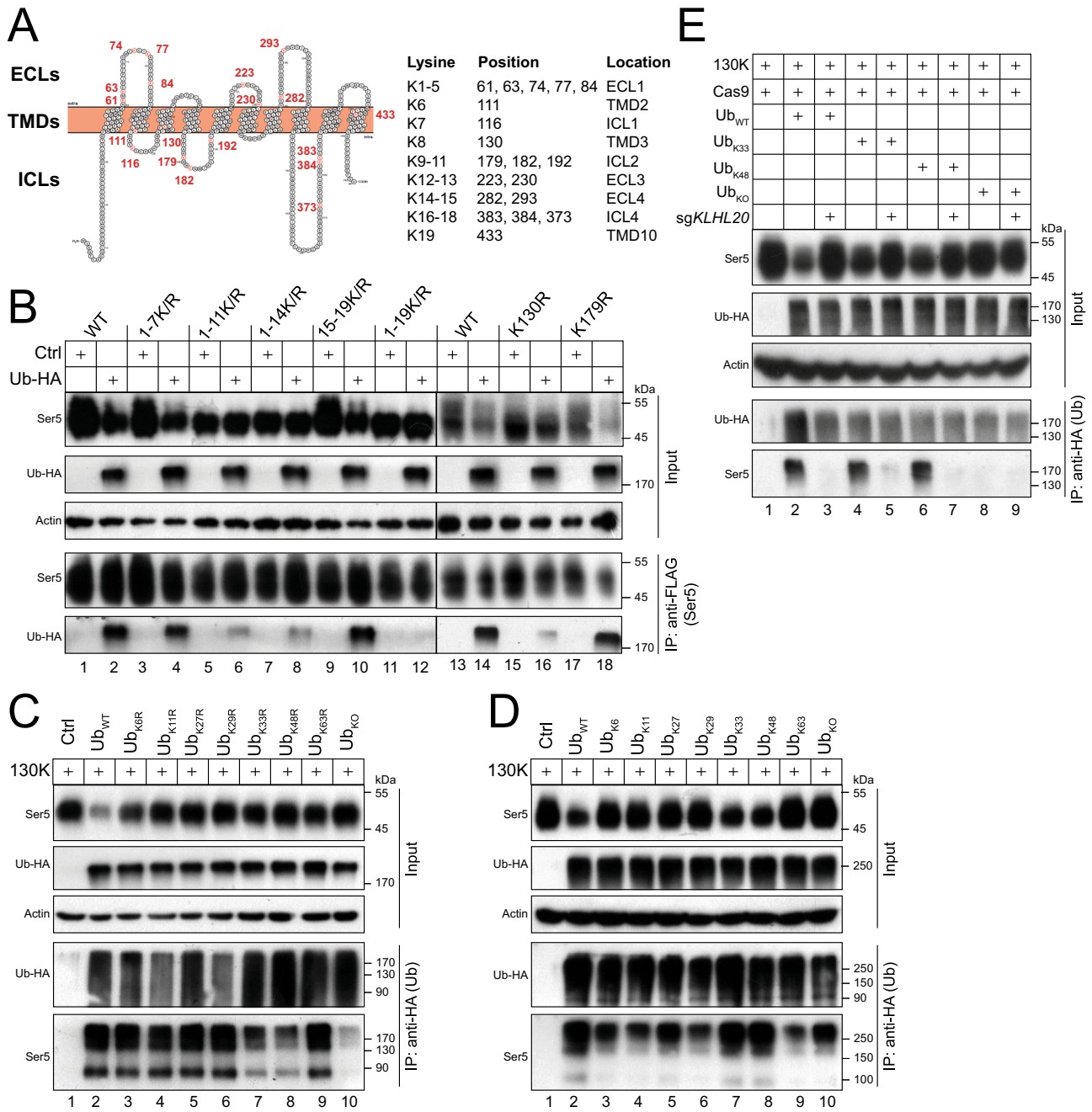

**Fig. 2 Identification of lysine 130 (K130) as a critical polyubiquitination site on SERINC5. A** A Ser5 membrane topology model is presented. Five ECLs, 10 TMDs, and 4 ICLs are indicated. Nineteen lysine residues (K1 to K19) are numbered and colored in red. Their distributions into these regions are indicated. Their positions in Ser5 protein are also indicated. The Ser5 membrane topology was predicted by the on-line transmembrane hidden Markov model (TMHMM, v2.0) (https://services.healthtech.dtu.dk/service.php?TMHMM-2.0) and the model was generated by Protter (http://wlab.ethz.ch/protter/start/). ECL, extracellular loop; TMD, transmembrane domain; ICL, intracellular loop. **B** Ser5 and its lysine mutants were expressed with Ub in HEK293T cells. Ser5 was immunoprecipitated by anti-FLAG and levels of polyubiquitinated Ser5 were compared by WB. **C** The Ser5-130K mutant (130 K) was expressed with Ub and its mutants bearing indicated K-to-R mutations in HEK293T cells. Ectopic Ub was immunoprecipitated by anti-HA and levels of polyubiquitinated Ser5 were compared by WB. **D** The 130 K mutant was expressed with indicated Ub mutants in HEK293T cells. Ectopic Ub was immunoprecipitated by anti-HA and levels of polyubiquitinated Ser5 were compared by WB. **E** The 130 K mutant was expressed with indicated Ub mutants in HEK293T cells in the presence or absence of *KLHL20*-KD. Ectopic Ub was immunoprecipitated by anti-HA and levels of polyubiquitinated Ser5 were compared by WB. All experiments were repeated at least twice, and similar results were obtained. Source data are provided as a Source Data file.

top panels). The 15-19 K/R mutant was localized to both the cell surface and the cytoplasm. The colocalization of WT, 1-7 K/R, 15-19 K/R, and K179R with DiIC$_{18}$(5), confirmed that these proteins localize to the plasma membrane (Fig. 3A, HeLa, bottom panels). In addition, when WT, K130R, and 1-19 K/R were expressed in human Jurkat T cells, only the WT protein was found on the cell surface, consistent with the results observed from HeLa cells (Fig. 3A, Jurkat).

Next, we analyzed SERINC5 expression on the cell surface by flow cytometry. A FLAG-tag was inserted into SERINC5

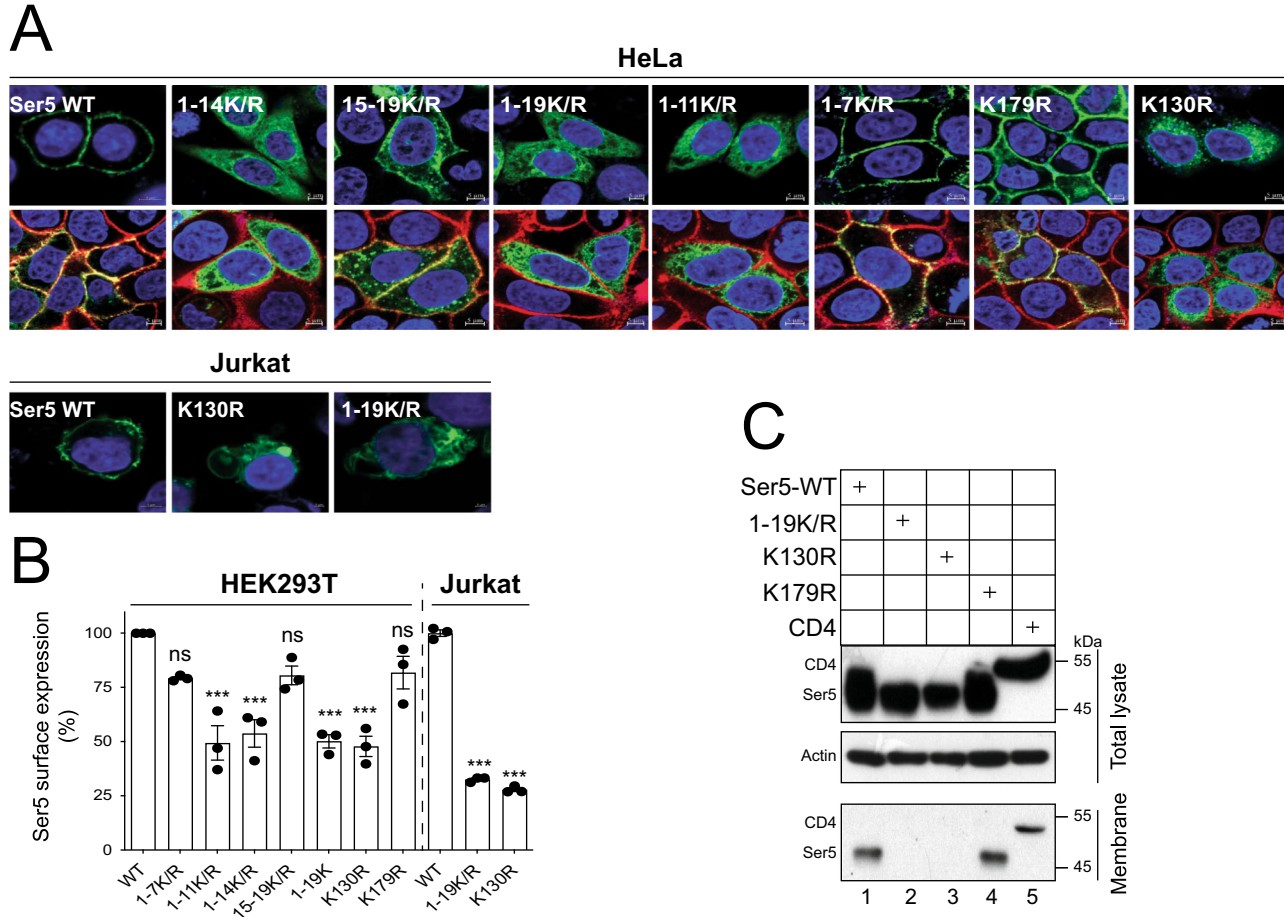

**Fig. 3 K130 is required for SERINC5 localization to the plasma membrane. A** Ser5-GFP and its lysine mutants were expressed in HeLa cells. Cells were stained with DAPI (4′,6-diamidino-2-phenylindole) for the nuclei (blue) and DilC$_{18}$(5) for the plasma membranes (far-red) (bottom panel) or only stained with DAPI (top panel). In addition, Ser5-GFP and its K130R or 1-19 K/R mutant were expressed in Jurkat cells, and cells were stained with DAPI. Subcellular localization of these Ser5 proteins was detected by confocal microscopy (scale bar 2 or 5 μm). Experiments were repeated twice, and representative experiments are shown. **B** Ser5-iFLAG and its lysine mutants were expressed in HEK293T and Jurkat cells. Cells were stained with anti-FLAG and levels of Ser5 on the cell surface were analyzed by flow cytometry. Results are shown as relative values, with the surface expression of WT Ser5 set as 100. Error bars indicate SEMs calculated from three experiments. $n = 3$; One-way ANOVA Tukey test; ns, not significant; *$p < 0.05$, **$p < 0.01$, ***$p < 0.001$. **C** Ser5, its lysine mutants, and CD4 were expressed in HEK293T cells. The plasma membranes from these cells were purified and analyzed by WB. Experiments were repeated twice, and similar results were obtained. Source data are provided as a Source Data file.

extracellular loop 3 region and similar lysine mutations were introduced into this SERINC5-iFLAG protein. These proteins were expressed in HEK293T cells, and their expression on the cell surface was determined by staining with anti-FLAG. SERINC5-iFLAG WT, 1-7 K/R, 15-19 K/R, and K179R were detected at much higher levels than 1-11 K/R, 1-14 K/R, 1-19 K/R, and K130R (Fig. 3B). When this experiment was repeated in Jurkat cells, the poor expression of 1-19 K/R and K130R on the cell surface was confirmed (Fig. 3B). These results are consistent with those from confocal microscopy.

Finally, we purified plasma membranes from cells to detect SERINC5 by WB. SERINC5 and its lysine mutants were expressed in HEK293T cells, and the plasma membrane fraction was isolated. CD4, a cell surface protein that is associated with the plasma membrane, was included as a control. Although WT SERINC5, 1-19 K/R, K130R, K179R, and CD4 were detected at a similar level in total cell lysate (Fig. 3C, total lysate), only the WT, K179R, and CD4 were found in the membrane fraction, whereas 1-19 K/R and K130R were not (Fig. 3C, membrane). Collectively, these results further confirm that K130 determines SERINC5 localization to the plasma membrane.

**Polyubiquitination is required for SERINC5 localization to the plasma membrane.** Although Cul3 and KLHL20 decreased SERINC5 expression in the presence of ectopic Ub (Fig. 1A), their ectopic expression and KDs did not affect SERINC5 expression at steady state (Fig. 4A). We then determined how Cul3 and KLHL20 affect SERINC5 subcellular localization by using the similar approaches. Notably, *Cul3-* or *KLHL20*-KD in HeLa and Jurkat cells reduced SERINC5 expression on the cell surface when detected by confocal microscopy (Fig. 4B) and flow cytometry (Fig. 4C, lanes 4, 6, 11, 13), and this reduction was rescued by ectopic Cul3 or KLHL20 expression (Fig. 4B, C, lanes 5, 7, 12, 14). In addition, these KDs also significantly reduced SERINC5 levels in the plasma membrane fraction (Fig. 4D, lanes 4, 6), which was also rescued by their ectopic expression (lanes 5, 7).

Next, we detected SERINC5 interaction with Ub in live cells via bimolecular fluorescence complementation (BiFC)[14–16]. In this assay, a basic yellow fluorescent protein Venus was divided into N-terminal (VN) and C-terminal (VC) fragments. HA-tagged VN and FLAG-tagged VC were fused to the C-terminus of Ub or SERINC5, respectively. When Ub-VN and SERINC5-VC were expressed together in HeLa cells, green fluorescence signals were

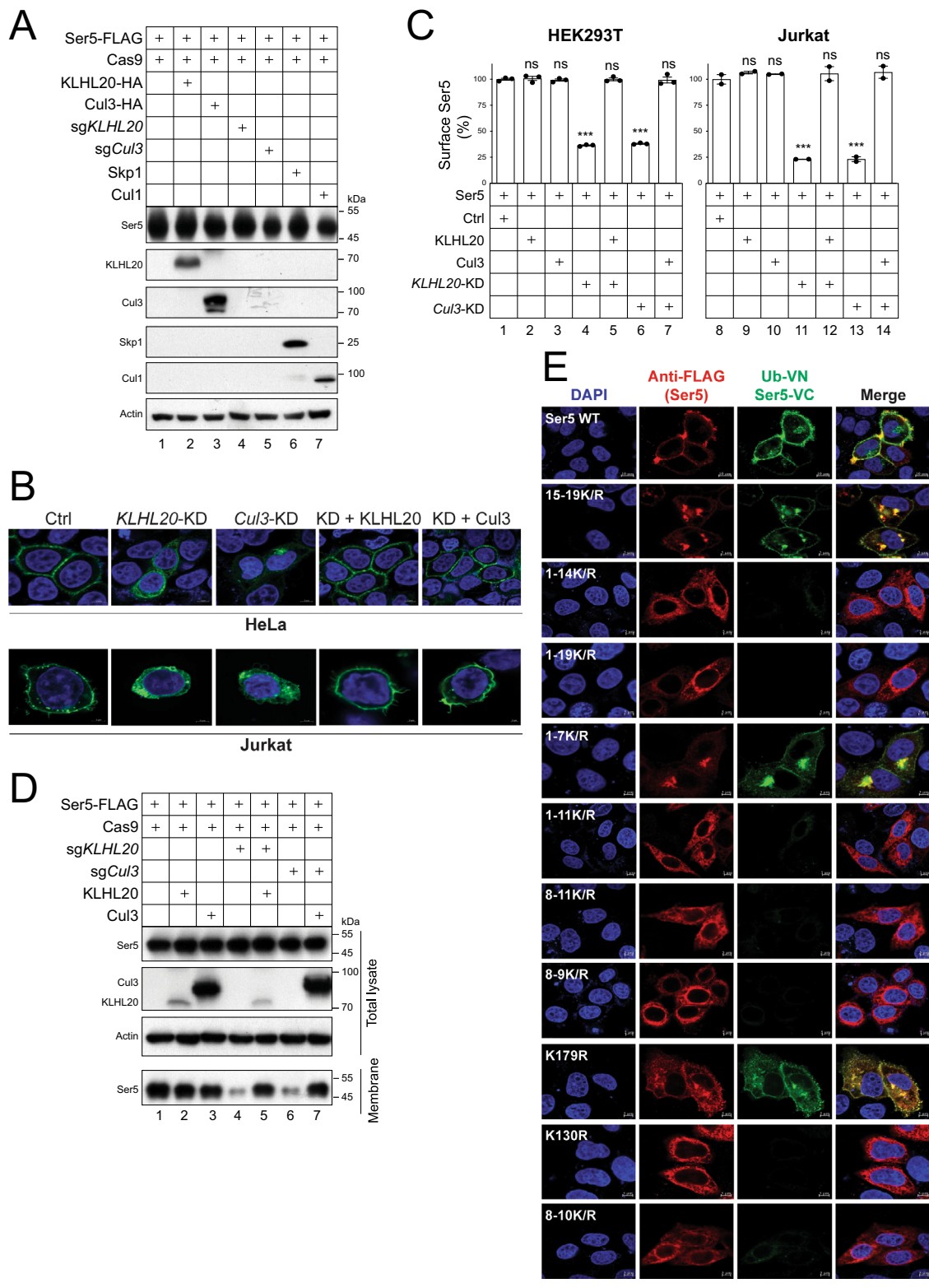

detected, indicating that Ub-SERINC5 interaction occurred in these cells (Fig. 4E). These BiFC signals co-localized with SERINC5, confirming the specificity of this interaction. Importantly, these signals primarily localized to the cell surface, consistent with the conclusion that ubiquitination is required for SERINC5's localization to the plasma membrane.

We then introduced the same lysine mutations into SERINC5-VC and tested how these mutants interact with Ub-VN. BiFC signals were detected in cells expressing 15-19 K/R, 1-7 K/R, and

K179R, but not 1-14 K/R, 1-19 K/R, 1-11 K/R, 8-11 K/R, 8-9 K/R, and K130R (Fig. 4E). In addition, these signals primarily localized to the cell surface. Collectively, these results confirm the important role of K130 polyubiquitination in SERINC5 localization to the plasma membrane.

**Polyubiquitination is required for SERINC5 downregulation by HIV-1 Nef.** To understand how SERINC5 polyubiquitination

**Fig. 4 Polyubiquitination is required for SERINC5 localization to the plasma membrane. A** Ser5 was expressed with Cas9, indicated proteins, and indicated sgRNAs in HEK293T cells. Ser5 expression was compared by WB. **B** Ser5-EGFP was expressed in HeLa and Jurkat cells in the presence of *KLHL20-* or *Cul3*-KD by CRISPR/Cas9. KLHL20 and Cul3 in these KD cells were also complemented by their ectopic expression. After staining with DAPI, Ser5 subcellular localization was detected by confocal microscopy (scale bar 2 or 5 μm). Experiments were repeated twice, and representative experiments are shown. **C** Ser5-iFLAG was expressed in HEK293T and Jurkat cells in the presence of ectopic KLHL20 and Cul3 expression and/or their KDs by CRISPR/Cas9. Cells were stained with anti-FLAG and levels of Ser5 on the cell surface were analyzed by flow cytometry. Results are shown as relative values, with the surface expression of Ser5 alone set as 100. Error bars indicate SEMs calculated from two or three experiments. $n = 3$ (HEK293T) or $n = 2$ (Jurkat); One-way ANOVA Tukey test; ns, not significant; *$p < 0.05$, **$p < 0.01$, ***$p < 0.001$. **D** Ser5 was expressed in HEK293T cells in the presence of ectopic KLHL20 and Cul3 expression and/or their KDs by CRISPR/Cas9. The plasma membranes from these cells were purified and analyzed by WB. **E** Ub-VN was expressed with Ser5-VC or its lysine mutants in HeLa cells. Cells were stained with DAPI and anti-FLAG for Ser5. BiFC fluorescent signals were detected by confocal microscopy (scale bar 2 or 5 μm). Experiments were repeated twice, and representative experiments are shown. Source data are provided as a Source Data file.

affects its antagonism by Nef, we expressed SERINC5 and its lysine mutants with Nef in HEK293T cells and analyzed SERINC5 downregulation by WB. Initially, we analyzed 9 lysine mutants and found that 1-14 K/R, 1-19 K/R, 1-11 K/R, 7-11 K/R, 7-14 K/R, 8-9 K/R, and 8-11 K/R were resistant to Nef (Fig. 5A, lanes 3–4, 7–8, 13–16, 19–20, 23–24, 31–34), whereas 15-19 K/R, 1-7 K/R, and 11-14 K/R were as sensitive to Nef as WT SERINC5 (lanes 5–6, 11–12, 21–22). These results suggested that K8 (K130) and K9 (K179) residues should be required for SERINC5 downregulation by Nef. We then directly tested K130R and K179R and found that K179R was still sensitive, whereas K130R became resistant to Nef (lanes 27–30). These results demonstrate that K130 determines SERINC5 sensitivity to Nef.

Previously, we reported that ectopic Ub synergizes SERINC5 downregulation by Nef[16]. We determined how these SERINC5 lysine residues affect such synergy. When SERINC5 proteins were expressed with Nef and Ub in HEK293T cells, Ub strongly promoted Nef downregulation of WT SERINC5 (Fig. 5B, lanes 1–4, 13–16). This synergy was also detected from Nef downregulation of the K179R mutant (lanes 21–24), but not the mutants containing the K130R mutation, such as 1-19 K/R, 8-11 K/R, and K130R (lanes 5–12, 17–20). These results further confirm that K130 is critical for SERINC5 downregulation by Nef.

To understand how K130 determines the SERINC5 sensitivity to Nef, we analyzed the SERINC5-Nef interaction in live cells via BiFC as we and others reported[16,24,25]. When HA-tagged VN and FLAG-tagged VC were fused to the C-terminus of SERINC5 or Nef and expressed in HeLa cells, BiFC signals were detected, and co-localized with SERINC5, indicating an association between SERINC5 and Nef in these cells (Fig. 5C). In addition, these signals were primarily detected in the cytoplasm, confirming Nef downregulation of SERINC5 from the cell surface. When lysine mutations were introduced into SERINC5-VN, Nef interacted with mutants 15-19 K/R, 1-7 K/R, and K179R, but not 1-14 K/R, 1-19 K/R, 1-11 K/R, 8-11 K/R, 8-9 K/R, and K130R (Fig. 5C). These results demonstrate that K130 is required for SERINC5 interaction with Nef in cells and suggest that SERINC5 polyubiquitination plays an indispensable role in this interaction.

**Polyubiquitination is required for SERINC5 anti-HIV-1 activity.** To understand how lysine residues affect SERINC5 incorporation into virions, WT and ΔNef HIV-1 were produced from HEK293T cells in the presence of SERINC5 and its lysine mutants. Virions were purified from culture supernatants by ultracentrifugation and analyzed by WB. We again confirmed that Nef downregulates 1-7 K/R, 15-19 K/R, and K179R, but not 1-11 K/R, 1-14 K/R, 1-19 K/R, and K130R in cells (Fig. 6A). Consistently, 1-7 K/R, 15-19 K/R, and K179R were detected in virions, and their incorporation was inhibited by Nef. In contrast, none of 1-11 K/R, 1-14 K/R, 1-19 K/R, and K130R were detected in virions. Thus, K130 is required for SERINC5 incorporation

into virions, which confirms its role in SERINC5 expression on the cell surface.

To further confirm the important role of polyubiquitination in SERINC5 antagonism by Nef, SERINC5 was expressed with HIV-1 Nef in HEK293T cells in the presence of *KLHL20-* or *Cul3*-KD by CRISPR/Cas9, and/or their ectopic expression. Nef effectively decreased SERINC5 expression (Fig. 6B, lanes 1-2), a phenotype which was further enhanced by ectopic KLHL20 and Cul3 expression (lanes 3–4). This decrease was disrupted by *KLHL20-* or *Cul3*-KD (lanes 6, 9), but restored upon complementation with their ectopic expression (lanes 7, 10). Thus, Nef downregulation of SERINC5 is dependent on Cul3-KLHL20 mediated SERINC5 polyubiquitination.

Finally, we determined how K130 and polyubiquitination affects SERINC5 anti-HIV-1 activity. Initially, ΔNef HIV-1 was produced from HEK293T and Jurkat cells in the presence of SERINC5 and its lysine mutants, and viral infectivity was analyzed after infection of HIV-1 luciferase-reporter TZM-bI cells. 1-7 K/R, 15-19 K/R, and K179R inhibited the viral replication as strongly as WT SERINC5, whereas 1-11 K/R, 1-14 K/R, 1-19 K/R, and K130R did not in both cell lines (Fig. 6C). Next, ΔNef HIV-1 was also produced from HEK293T and Jurkat cells in the presence of SERINC5 and *KLHL20-* or *Cul3*-KD, and viral infectivity was analyzed again as above. *KLHL20-* or *Cul3*-KD disrupted the SERINC5 antiviral activity in both cell lines, a phenotype which was restored upon complementation with their ectopic expression (Fig. 6D, lanes 5–8). Thus, K130 and polyubiquitination are required for SERINC5 anti-HIV-1 activity.

**Discussion**
KLHL proteins normally have three functional domains: BTB, BACK, and Kelch. The BACK domain bridges the BTB domain with the Kelch domain and it also has a N-terminal structural motif called a 3-box that forms a cleft. In combination with the C-terminus of the BTB domain, the 3-box motif binds to the N-terminal tail of Cul3[26]. The Kelch domain has six Kelch repeats that serve as the substrate recognition domain, and this domain from KLHL20 binds to death-associated protein kinase (DAPK)[23]. We now show that KLHL20 binds to SERINC5 via the Kelch domain and recruits SERINC5 to CRL3^KLHL20 for being polyubiquitinated, which potentiates the SERINC5 post-Golgi trafficking to the plasma membrane and regulates its expression.

TGN is a major soring station in the secretory pathway that is responsible for expression of secretory proteins and integral membrane proteins. It has been unclear how the SERINC5 anterograde trafficking from the Golgi to the plasma membrane is potentiated, which is a critical process for SERINC5 to exhibit its antiviral activity. We now show that this process is dependent on SERINC5 polyubiquitination at K130 by CRL3^KLHL20. A recent study on SERINC5 protein structure suggests that this K130 residue is located closely to the inner boundary of the plasma

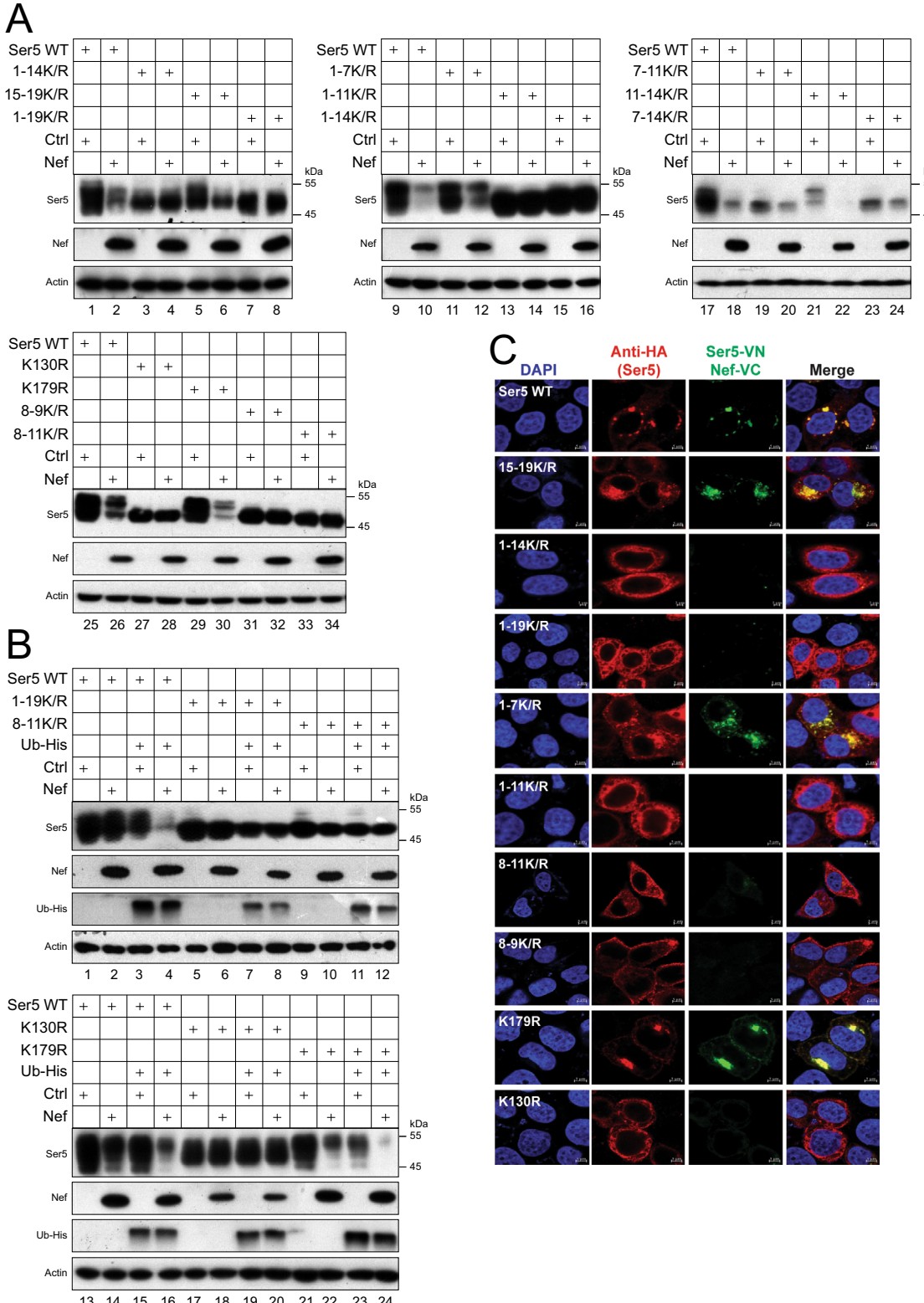

**Fig. 5 Polyubiquitination is required for SERINC5 downregulation by HIV-1 Nef. A** Ser5 and its lysine mutants were expressed with HIV-1 Nef in HEK293T cells, and their expression was analyzed by WB. Ctrl, control vector. **B** Ser5 and its lysine mutants were expressed with Ub and Nef in HEK293T cells, and their expression was analyzed by WB. **C** Ser5-VN and its lysine mutants were expressed with Nef-VC in HeLa cells. Cells were stained with DAPI and anti-HA for Ser5, and BiFC fluorescent signals were detected by confocal microscopy (scale bar 2 or 5 μm). Experiments were repeated twice, and representative experiments are shown. Source data are provided as a Source Data file.

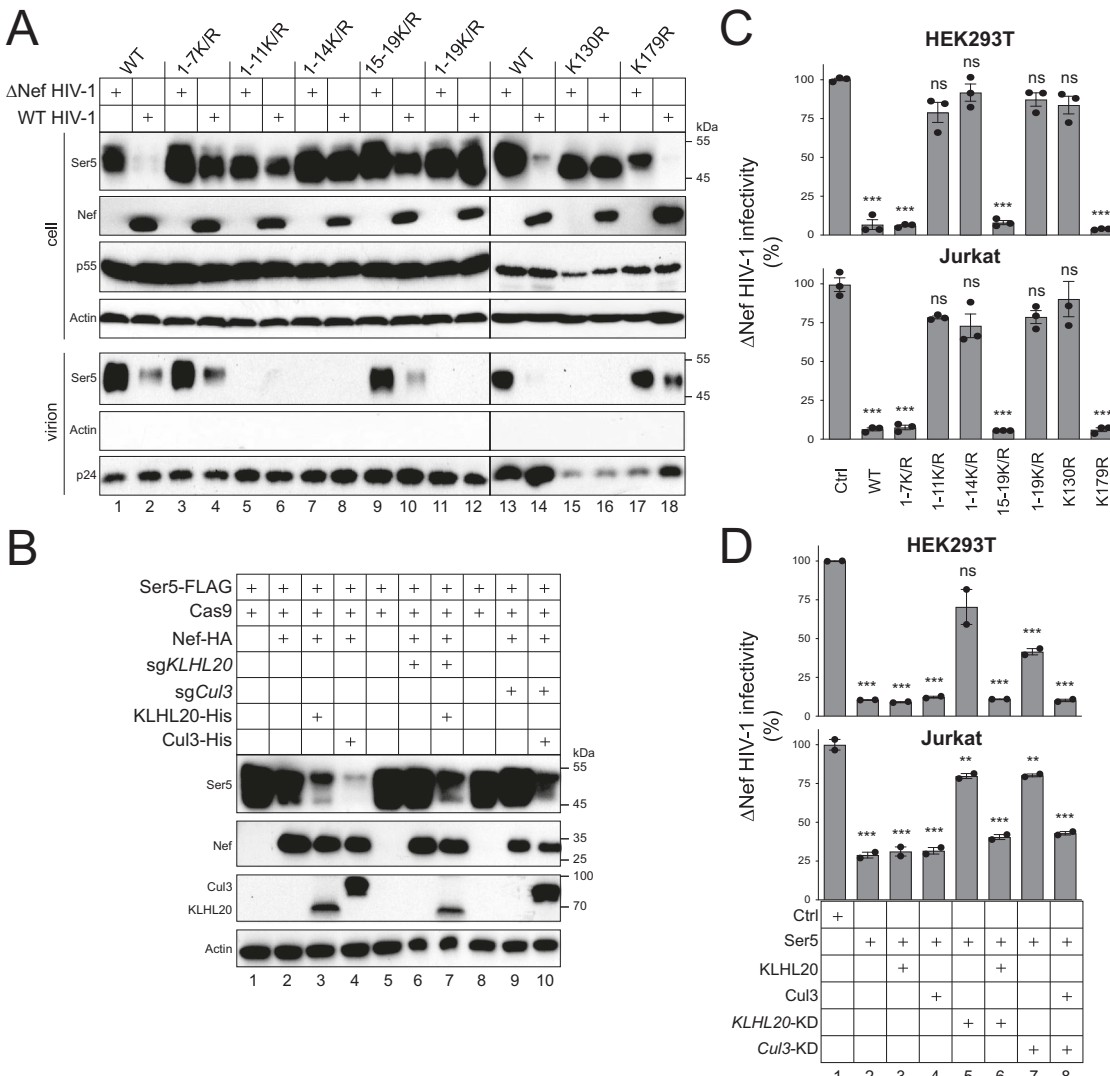

**Fig. 6 Polyubiquitination is required for SERINC5 anti-HIV-1 activity. A** WT and ΔNef HIV-1 were produced from HEK293T cells in the presence of Ser5 and its lysine mutants. Virions were purified from culture supernatants via ultracentrifugation. Ser5 expression in cell lysate and virions were detected by WB. **B** Ser5 was expressed with Nef in HEK293T cells in the presence of ectopic KLHL20 and Cul3 expression and/or their KDs by CRISPR/Cas9. Ser5 expression was compared by WB. **C** ΔNef HIV-1 was produced from HEK293T and Jurkat cells in the presence of Ser5 and its lysine mutants. After normalization by p24$^{Gag}$ ELISA, viral infectivity was analyzed after infection of HIV-1 luciferase reporter cell line TZM-bl. **D** ΔNef HIV-1 was produced from HEK293T and Jurkat cells in the presence of SERINC5, ectopic KLHL20 and Cul3 expression, and/or their KDs by CRISPR/Cas9. Viral infectivity was analyzed similarly as in (**C**). For **C** and **D**, results are presented as relative values, with the infectivity of viruses produced in the presence of a control vector (Ctrl) vector set as 100. Error bars indicate SEMs calculated from two or three experiments. $n = 3$ (**C**) or $n = 2$ (**D**); One-way ANOVA Tukey test; ns, not significant; *$p < 0.05$, **$p < 0.01$, ***$p < 0.001$. All experiments were repeated twice, and representative experiments are shown. Source data are provided as a Source Data file.

membrane[4]. The long side chain of this lysine should be able to stretch out of the membrane interior via "snorkeling", which exposes its positively charged amino group to the cytoplasm for ubiquitination, as demonstrated in other studies[27,28]. We also show that the K130R mutant does not traffic to the cell surface and does not have any antiviral activity. We have multiple pieces of evidence to confirm that this mutant is not misfolded/aggregated in cells, which otherwise could also result in its mislocalization and loss of function. First, we found that K130R and 1-19 K/R still oligomerized as the WT SERINC5 when detected by BiFC (Supplementary Fig. 1A). Second, we found that K130R 1-19 K/R also interacted with KLHL20 but not its ΔK mutant (Supplementary Fig. 1B). Third, we found that K130R and 1-19 K/R retained the SERINC5 activity that potentiates the

NF-κB signaling pathway (Supplementary Fig. 1C), as reported recently[29]. The same paper also reported that SERINC5 interacts with the TNF Receptor Associated Factor 6 (TRAF6) during the potentiation. Consistently, K130R and 1-19 K/R also interacted with TRAF6 (Supplementary Fig. 1B). Collectively, these results demonstrate that our K-to-R SERINC5 mutants are properly folded and still active. Thus, K130 polyubiquitination is very critical for SERINC5 expression on the cell surface.

Notably, we found that SERINC5 is polyubiquitinated at K130 via both K33- and K48-linked Ub chains. Although Ub chains can be homotypic that are linked through the same acceptor site of ubiquitin, they are also heterotypic that are linked through multiple sites. Heterotypic chains are further divided into a mixed single chain or at least two branched chains that have two

linkages within the same polymer[30]. So far, four types of branched Ub chains that have clear physiological functions have been identified, including K11/K48[31], K29/K48[32], K48/K63[33], K63/M1[34]. Several other branched Ub chains with unidentified functions have also been detected. In fact, up to 20% Ub chains detected from cellular proteins are branched chains, indicating that they play a critical role in cellular functions. Branched chains can be assembled via at least three mechanisms by (1) paired E3s that have distinct linkage specificities; (2) a single E3 with paired E2s that have distinct linkage specificities; (3) a single E3 with a single E2. Our results strongly suggest that K33/K48-linked chains are assembled during polyubiquitination, which has not been described in any proteins so far to the best of our knowledge. These results further suggest that, in addition to homotypic K33- or K48-linked chain, CRL3^KLHL20 also generates K33/K48-linked chains. CRL3^KLHL20 generates K48-linked chains with different E2s, but its E2 for K33-linked chain assembly is still unknown. Thus, the mechanism of how CRL3^KLHL20 generates K33/K48-linked chains should be further investigated.

Our results further suggest that K33/K48-linked chains have dual functions, which is distinctive from the single function of the other branched chains (Fig. 7). K11/K48- and K29/48-branched chains act as a more powerful degradation signal and promote degradation in the proteasomes[31,32], whereas K29/K48- and K63/M1-branched chains serve as nondegradable signals and contribute to cell signaling events[33,34]. In the case of SERINC5 polyubiquitination, K33-linked chains should serve as a sorting signal that transports SERINC5 from TGN to the plasma membrane. K33-linked chains may interact with a Ub-binding protein such as Eps15 that is required for the post-Golgi trafficking, as implicated in the coronin 7 signaling cascade[35]. Once SERINC5 is localized to the plasma membrane, K48-linked chains should sort SERINC5 to the lysosomes via endocytic pathway for degradation when Nef is present. Although the K48-linked chain generated by CRL3^KLHL20 has been found to promote proteasomal degradation, it is clear that K48-linked chains also target proteins to lysosomes for degradation[36]. Collectively, K33/K48-linked chains should act as both nondegradable and degradable signals that potentiate and regulate SERINC5 expression on the cell surface.

It would be very interesting if K33/K48-linked chains play a very general role in regulating cellular protein expression. It is still unclear how expression and trafficking of integral membrane proteins are regulated at post-Golgi levels. However, transmembrane proteins are often downregulated from the plasma membrane via endocytic pathway for degradation. Thus, their expression must be tightly controlled at cellular levels by a negative mechanism. K33/K48-linked chains may act as a positive regulator by promoting their targeting to the plasma membrane via the K33-linked chain, and a negative regulator by promoting their downregulation from the plasma membrane via the K48-linked chain. It is noteworthy that HIV-1 is very adept to take advantage of this negative regulatory mechanism and antagonize SERINC5 by accelerating its degradation by Nef (Fig. 7). Thus, further understanding the biology of K33/K48-linked chains will collect new insights into the general mechanism for integral cellular membrane expression and the arms race between SERINC5 and Nef during HIV-1 infection.

## Methods

**Cell Lines**. The human HEK293T cells and HeLa cells were obtained from the American Type Culture Collection. TZM-bI cells were obtained from the National Institutes of Health (NIH) HIV Reagent Program. HEK293T and TZM-bI cells were maintained in Dulbecco modified Eagle medium (DMEM) with 10% fetal bovine serum (Gibco), at 37 °C and 5% $CO_2$. The human Jurkat-TAg (JTAg) cell line was provided by Heinrich Gottlinger and cultured in RPMI 1640 with 10% fetal bovine serum (FBS) (Sigma), at 37 °C and 5% $CO_2$.

**Bacterial strains**. *Escherichia coli* HB101 (Promega) was used as the recipient strain for preparation of HIV-1 proviral vectors, whereas all the other plasmid vectors were prepared from *Escherichia coli* DH10α (Vazyme). All these bacteria were cultured in Luria-Bertani (LB) broth in a shaking incubator at 37 °C.

**Expression vectors**. The Env-deficient HIV-1 proviral vector pNLΔE (pNLen-CAT), its Nef-deficient version pNLΔEΔN (pNLenCAT-Xh), and HIV-1 Env expression vector pNLnΔBS were provided by Kenzo Tokunaga[37]. The following plasmids pCMV6-Ser5-FLAG, pCMV-Ser5-EGFP, pCMV-HA-Ub, pCMV-His₆-Ub, pcDNA3.1-Ser5-FLAG-VC, pcDNA3.1-Ser5-VN-HA, pcDNA3.1-Ub-VN-HA, pcDNA3.1-SF2Nef-V5-VC, pcDNA3.1-SF2Nef-HA, and pCMV6-CD4-FLAG were reported previously[14,16,19]. pLVXm-N-FLAG-Skp1 and pLVXm-N-FLAG-Cul1 were provided by Yan Chen[38]. The NF-κB-luciferase reporter vector was provided by Shan-Lu Liu. The TRAF6-HA expression vector was provided by Haitao Wen.

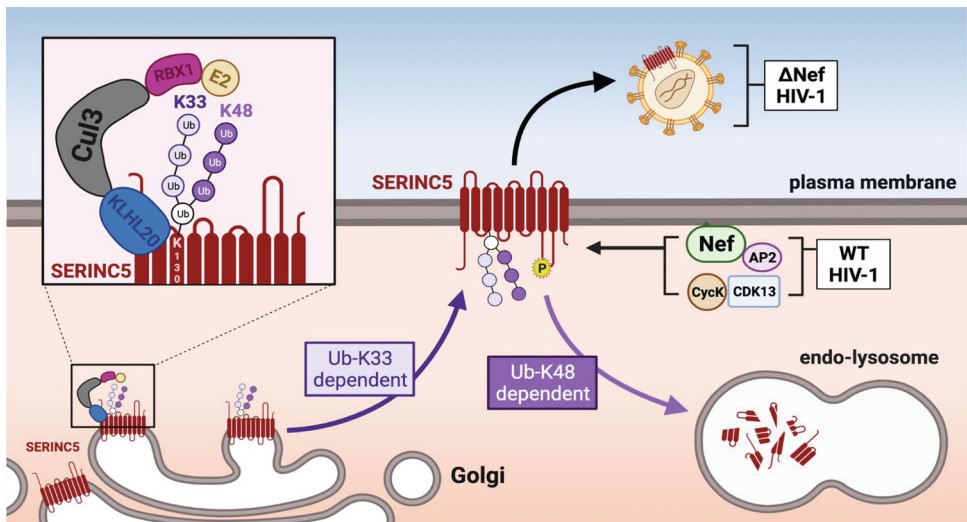

**Fig. 7 A model of how CRL^KLHL20 determines SERINC5 trafficking and expression.** CRL^KLHL20 catalyzes SERINC5 polyubiquitination at the TGN and produces K33/K48-brached chains at K130 of SERINC5. The K33-linked Ub chain (Ub-K33) potentiates SERINC5 trafficking to the plasma membrane, where it is packaged into HIV-1 virions to inhibit viral infection in the absence of Nef. However, when Nef is expressed, this viral accessory protein recruits CycK/CDK13 to phosphorylate the ICL4 of SERINC5. This phosphorylation triggers SERINC5 conformational change, resulting in recruitment of the AP2 complex to SERINC5 via Nef. SERINC5 is then targeted to the endocytic pathway and degraded in lysosomes via the K48-linked Ub chain (Ub-K48). BioRender was used for making the model.

pcDNA3.1-iFLAG-Ser5 that has a FLAG tag inserted between residues 290 and 291 of Ser5 was created by overlapping PCR amplification. pCMV6-Ser5 1-19 K/R-FLAG, pCMV6-KLHL20-HA, pCMV6-Rbx1-HA, and pCMV6-Cul3-HA were constructed from Comate Bioscience. The Kelch-repeat domain deletion mutant pCMV6-KLHL20-ΔK-HA (KLHL20ΔK: residues 1–317) were generated by PCR. The pCMV6-KLHL20-5A-HA mutant, in which the residues V110, R112, I114, L147, and Q149 were each replaced by alanine, was generated by site-directed mutagenesis. pCMV6-KLHL20-His$_6$, pCMV6-Rbx1-His$_6$, pCMV6-KLHL20-ΔK-His$_6$, pCMV6-KLHL20-5A-His$_6$, and pCMV6-Cul3-His$_6$ were generated by PCR. The Cas9 expression vector pMJ920 was obtained from Jennifer Doudna through Addgene. A *KLHL20 E2g1 gRNA* (5′-GCGAACGTTTAGCTCATCACTGG-3′) targeting the 2nd exon, *KLHL20 E1g2 gRNA* (5′- TGATCCGAGACATTGACGAGA GG -3′) targeting the 1st exon; and *Cul3 E1g1 gRNA* (5′- CGAGATCAAGTTGT ACGTTATGG -3′), *Cul3 E1g2 gRNA* (5′- GACCACTGTTATTCTTACGCTGG-3′) targeting the 1st exon, were expressed from pGEM-T (Promega) as we did before[39].

Ser5 1-14 K/R, 15-19 K/R, 1-19 K/R, 1-7 K/R, 1-11 K/R, 7-11 K/R, 11-14 K/R, 7-14 K/R, 8-9 K/R, 8-11 K/R, K130R, and K179R mutation in pCMV6-Ser5-FLAG, pCMV-Ser5-EGFP, pcDNA3.1-Ser5-VN-HA, pcDNA3.1-Ser5-VC-FLAG, and pcDNA3.1-iFLAG-Ser5 were created by a site-directed mutagenesis, respectively. Ser5 130 K mutation was created in pCMV6-Ser5-1-19K/R-FLAG vector. Primers and cloning methods are available upon request.

**Western blotting (WB)**. Typically, HEK293T cells were seeded either in 6-well plates or in 6-cm dishes with an initial density of $5 \times 10^5$ cells per well or $1 \times 10^6$ cells per dish. After 24 h or 48 h of transfection, virions were purified from culture supernatants by ultracentrifugation. In addition, membrane proteins were extracted from cells by Membrane and Cytosol Protein Extraction Kit (Beyotime). Cells were lysed with RIPA buffer (25 mM Tris, pH 7.4, 150 mM NaCl, 0.5% sodium deoxycholate, 0.1% SDS, 1% Nonidet P-40) supplemented with protease inhibitor cocktail (Sigma-Aldrich, P8340). Proteins in cytosol, virions, and membrane faction were resolved by sodium dodecyl sulfate polyacrylamide gel electrophoresis (SDS-PAGE). Separated proteins were transferred onto polyvinylidene difluoride (PVDF) membranes and membranes were blocked with 5% nonfat milk powder in TBST (Tris-buffered saline [20 mM Tris, pH 7.4, 150 mM NaCl] containing 0.1% Tween 20) for 1 h at room temperature. Membranes were then probed by primary antibody followed by HRP-conjugated secondary antibodies. Primary and secondary antibodies were used at 1,000 or 2,000 dilutions, respectively. A list of these antibodies is provided in Reporting Summary in Supplementary information. Chemiluminescence signals were then measured by incubating the membrane with SuperSignal substrate (Thermo Fisher Scientific). Adobe Photoshop 2021 was used to analyze western blot images. Adobe Illustrator 2021 was used to create figures.

**Immunoprecipitation (IP)**. To detect protein interactions in Figs. 1F, G and 2C–E HA-tagged proteins were expressed with their target proteins in HEK293T cells cultured in 6-cm dishes. Proteins were pulled down by an Anti-HA-Agarose antibody and analyzed by WB. To detect protein interactions in Figs. 1D, E and 2B, FLAG-tagged proteins were expressed in HEK293T cells cultured in 6-cm dishes. Proteins were pulled down by an anti-FLAG M2 antibody and analyzed by WB.

**Viral infectivity**. Viruses were produced from HEK293T and Jurkat-TAg cells after transfection as described previously[16]. In brief, Ser5 and its indicated lysine mutant expression vector in the presence or absence of silencing vectors or expression vectors (KLHL20, Cul3) were transfected either in HEK293T or Jurkat-TAg cells. After 48 h, viruses were collected from the culture supernatants, and viral production was quantified by p24$^{Gag}$ ELISA. To determine viral infectivity, equal amounts of viruses were used to infect the HIV-1 luciferase reporter cell line TZM-bI in a 96-well plate at a density of $1 \times 10^4$ per well. After 48 h, cells were lysed, and intracellular luciferase activities were determined using the Bright-Glo Luciferase Assay System (Promega).

**Flow cytometry**. The HEK293T and Jurkat-TAg cells were transfected with Ser5 and its indicated lysine mutant expression vectors for 24 h. The cells were then fixed with 4% paraformaldehyde for 5 min and blocked with 5% BSA for 2 h. The cells were incubated with an anti-FLAG monoclonal antibody at 1,000 dilutions at 4 °C overnight. After washing, the cells were incubated with Alexa Fluor 488-conjugated goat anti-mouse antibody at 2,000 dilutions for 1 h. Ser5 expression on the cell surface was determined by flow cytometry. BD FACSDiva (v7.0) and FlowJo (v10.8) were used for data analysis. The gating strategies for HEK293T and Jurkat cells are shown in Supplementary Fig. 2.

**Confocal microscopy**. A total of 1.5~2.0 × 10$^5$ HeLa cells were seeded in coverslips and transfected with indicated vectors using Lipofectamine® 3000 as a transfection reagent. After 24 h, cells were fixed with 4% paraformaldehyde for 5 min and permeabilized with 0.1 % Triton X-100 for 10 min, and then blocked with 10% FBS for 2 h. For immunofluorescence assay, cells were incubated with anti-HA or anti-FLAG antibodies overnight at 4 °C. After washing, the cells were incubated with Alexa Fluor 488-conjugated goat anti-mouse antibody for 1 h, and then incubated with 4′,6-diamidino-2-phenylindole (DAPI) (Sigma) for 30 s for nuclear staining. Finally, cells were observed under confocal microscope (ZEISS, LSM880). The

Jurkat-TAg cells were prepared by centrifugation at $400 \times g$ for 5 min at 4 °C. After fixation, permeabilization, blocking and staining cells with DAPI, cells were mounted on a glass-slide and covered by a coverslip. Slides were left in dark at room temperature for 30 min before they were examined under Zeiss LSM 880 confocal microscope.

**Statistical analysis**. All experiments were performed independently at least two times. SPSS Statistics Software (Version 23; IBM, Inc., New York, USA) was used for the data analysis. Quantitative values of data were expressed as mean ± standard error of measurements (SEMs) and represented by error bars. Comparisons were analyzed by one-way analysis of variance (ANOVA) followed by Tukey test. A $p$ value < 0.05 ($p < 0.05$) was considered to be statistically significant when *$p < 0.05$, **$p < 0.01$, ***$p < 0.001$, ns (not significant, $p > 0.05$).

**Reporting summary**. Further information on research design is available in the Nature Research Reporting Summary linked to this article.

## Data availability
Source data are provided with this paper.

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

## Acknowledgements
We thank Heinrich Gottlinger, Ruey-Hwa Chen, Shan-Lu Liu, Haitao Wen, and the NIH HIV Reagent Program for providing reagents. The Fig. 7 model was created with BioRender.com. S.L. was supported by a grant from National Natural Science Foundation of China (32172836). Y.H.Z. is supported by a grant from National Institutes of Health (AI145504).

## Author contributions
R.L., I.A., X.L. and S.L. performed all experiments except mass spectrometry analyses. L.S. conducted mass spectrometry analysis. S.F.J. created the model and provided insightful comments on the paper. S.L. and Y.H.Z. designed this study. Y.H.Z. wrote manuscript with input from all coauthors.

## Competing interests
The authors declare no competing interests.
