## [Peer Review File · Nature Communications]

Reviewer comments, first round –

Reviewer #1 (Remarks to the Author):

In this manuscript, Li and collaborators provide mechanistic details of how SERINC5 expression at the plasma membrane is modulated. Specifically, they found that the E3 ubiquitin ligase Cullin 3-KLHL20 polyubiquitinates SERINC5 at Lys 130 via K48 and K33-linked Ub chains. The authors further found that this is the main SERINC5 isoform present at the plasma membrane and antagonized by Nef. Also, this form is responsible for the SERINC5-dependent infectivity defect of HIV. Overall, this work fills important gaps in our understanding of how SERINC5 restricts HIV and the mechanism by which Nef counteracts it. The data presented is solid, supported by multiple complementary experiments. I only have minor suggestions

Fig. 1F and lines 124-127. The concept illustrated in the text is not presented in the figure. Rather than cropping the membranes, the authors should show full membranes for SERINC5 to see whether higher molecular forms of SERINC5 (ubiquitinated at higher than 90 kDa) can be observed

Fig. 2. The authors should provide names to their mutants (K1-5 etc) in a way that is more intuitive to locate the residues being mutated

The authors conclude that SERINC5 is ubiquitinated at K130 by multibranch K33/K48 Ub chains. Since this has never been reported for other proteins, how do the authors know that this dual positive result is not due to the fact that SERINC5 K130 can either become ubiquitinated by K33-poly-Ub chains or K48-poly-Ub chains, and that both types of post-translational modifications cause a similar phenotype?

Fig. 6A. The authors should show purity of the pelleted virion fraction by showing that actin is not detected in those samples.

Finally, I would recommend the authors to have the text revised by a native English speaker. There are multiple sentences with grammatical mistakes

Reviewer #2 (Remarks to the Author):

The manuscript describes the potential role of ubiquitination on correct transport, antiviral activity and sensitivity to HIV-1 Nef of SERINC5. The authors identify Lysine 130 as the main ubiquitination site of SERINC5 mediated by Cullin3. The work is original and stems from previous results of the Zheng group, which identified interactors of SERINC5 by Mass Spec. The results described here are potentially very interesting and provide a possible mechanism that regulates SERINC5 expression and activity. However, in my opinion, there are two major concerns that should be addressed.

1) K130 is identified as the site of SERINC5 ubiquitination. However, the structure of SERINC is now available, as determined by cryoEM, and shows that K130 is part of an alpha helix embedded within the membrane. This would suggest that the residue should not be accessible for ubiquitination. It is therefore very difficult to reconcile the results of this study with the conserved conformation of the protein. There is a real possibility that the K130R mutant could be misfolded and therefore what is observed here could be the effect of aberrant conformation of the mutant. On the same line, the failure of the K130R mutant to localize on the cell membrane and be incorporated into virion particles could be the consequence of the misfolding of the protein. Similarly, the failure of the mutant to be targeted by Nef could have the same explanation. Direct evidence that K130 is ubiquitinated should therefore be produced to rule out artefacts. One way to demonstrate direct ubiquitination is by identifying diGly peptides by Mass Spectrometry, which I suggest.

2) In general, all experiments performed to test SERINC5 degradation and poly-ubiquitination are performed without a proper control protein. With the current data, it is difficult to understand whether overexpression of Ub and components of the Ub ligases complexes have a generalized and non-specific degradative effect on proteins with available lysines. The only control shown, in fact, is a mutant SERINC with no lysines, which again may be totally misfolded and in any case is intrinsically resistant to ubiquitination. A non-SERINC5 membrane protein would therefore be desirable to assess if these experiments, where proteins are ectopically overexpressed, are

revealing a true physiological effect.

Reviewer #1 (Remarks to the Author):

I only have minor suggestions

Fig. 1F and lines 124-127. The concept illustrated in the text is not presented in the figure. Rather than cropping the membranes, the authors should show full membranes for SERINC5 to see whether higher molecular forms of SERINC5 (ubiquitinated at higher than 90 kDa) can be observed

The full membrane for SERINC5 from IP is now presented (**Fig.1F**, bottom panel). Much more SERINC5 proteins at higher than 90 kDa were clearly detected in the presence KLHL20 or Cul3. We did not detect these polyubiquitinated SERINC5 proteins from the input samples.

Fig. 2. The authors should provide names to their mutants (K1-5 etc) in a way that is more intuitive to locate the residues being mutated

Thanks for this suggestion. The positions of 19 lysine residues in SERINC5 are fully annotated (**Fig.2A**), which should make it easier to locate the residues being mutated.

The authors conclude that SERINC5 is ubiquitinated at K130 by multibranch K33/K48 Ub chains. Since this has never been reported for other proteins, how do the authors know that this dual positive result is not due to the fact that SERINC5 K130 can either become ubiquitinated by K33-poly-Ub chains or K48-poly-Ub chains, and that both types of post-translational modifications cause a similar phenotype?

We showed that Cul3/KLHL20 polyubiquitinates the K130 of SERINC5 via K33-linked and K48-linked ubiquitin chains with a similar efficiency (**Fig.2C, 2D, 2E**). These results suggest that the K33 and K48 polyubiquitination should occur concurrently but not independently, resulting in branched K33/K48 chains but not separated K33 and K48 chains. Currently, we are not aware of any evidence to suggest that two different ubiquitin chains can be formed separately on the same lysine residue during polyubiquitination.

Fig. 6A. The authors should show purity of the pelleted virion fraction by showing that actin is not detected in those samples.

Thanks for this suggestion. We now show that actin is not detected in virion samples (Fig.6A).

Finally, I would recommend the authors to have the text revised by a native English speaker. There are multiple sentences with grammatical mistakes

The English has been further edited.

Reviewer #2 (Remarks to the Author):

However, in my opinion, there are two major concerns that should be addressed.

1) K130 is identified as the site of SERINC5 ubiquitination. However, the structure of SERINC is now available, as determined by cryoEM, and shows that K130 is part of an alpha helix embedded within the membrane. This would suggest that the residue should not be accessible for ubiquitination. It is therefore very difficult to reconcile the results of this study with the conserved conformation of the protein.

In the current SERINC5 structural model, K130 is localized closely to the inner boundary of the plasma membrane. It is known in the membrane protein field, that the long side chain of lysine in a protein transmembrane domain can stretch out of the membrane interior via a “snorkeling” mechanism (PMIDs: 12782292, 28007883, 22734656, 27601675). Accordingly, the positively charged amino group of K130 should be exposed to the cytoplasm.

In fact, we now have evidence that this K130 residue itself is exposed to the cytoplasm. As suggested by this reviewer (see below), we analyzed Ser5 proteins from cells. We purified Ser5 from HEK293T cells by anti-FLAG column and after digestion of Ser5 with trypsin, samples were subjected to Mass Spectrometry analysis. In total, we detected 18 Ser5 peptides: 8 from ECL1, 8 from ICL4, 1 from ICL1, and 1 from ICL2 (Fig.S2). We could not detect any peptides from its transmembrane domains. We found one from ICL1 (¹²⁰AHIHNGFWFFK¹³⁰) that contains K130. The fact that trypsin cleaves Ser5 at K130 demonstrates that this residue is not protected by the membrane. Otherwise, we should not detect this peptide by this method. Thus, K130 is exposed and accessible for ubiquitination in cells.

A summary of Ser5 peptides detected from mass spectrometry analysis from seven independent experiments

There is a real possibility that the K130R mutant could be misfolded and therefore what is observed here could be the effect of aberrant conformation of the mutant. On the same line, the failure of the K130R mutant to localize on the cell membrane and be incorporated into virion particles could be the consequence of the misfolding of the protein. Similarly, the failure of the mutant to be targeted by Nef could have the same explanation.

Lysine and arginine are basic amino acids that share an almost identical structure. This fact means that K-to-R substitutions rarely affect protein conformation. Therefore, K-to-R mutagenesis is a technique that has been widely used for decades, with great success, to study protein ubiquitination and other protein functions. The speculation that a single K-to-R mutation, out of the 423 total amino acids that make up Ser5, causes protein misfolding or aggregation is, in our opinion, an extreme position to take. That being said, in response to this speculation, we present three new experiments that demonstrate that not only the K130R mutant, but also the 1-19K/R mutant, which has all 19 lysine residues replaced with arginine, are both functionally active.

First, we used bimolecular fluorescence complementation (BiFC) to detect protein misfolding and aggregation. This assay is like those split GFP or luciferase assays that have been widely used for this type of detection (PMID: 34246835, 23565019). We and others already reported that Ser5 oligomerization can be detected by BiFC. However, if Ser5 is aggregated or insoluble, it will not interact with each other, therefore does not produce BiFC fluorescence signals. We found that both K130R and 1-19K/R can oligomerize as effectively as the WT Ser5 (**Fig.S1A**). Consistently, the green fluorescent BiFC signals from the WT proteins were detected from the cell surface, and those from these two mutants were detected in the cytoplasm, as we already showed in this manuscript.

Second, we compared how these Ser5 proteins interact with KLHL20. We showed that Ser5 interacts with KLHL20 that is responsible for the assembly of the Cul3-KLHL20 E3 ubiquitin ligase (Fig.1E, Fig.1G). We now show that the K130R and 1-19K/R also bind to KLHL20 but not its Δ K mutant (**Fig.S1B**, lanes 1-6).

Third, we directly compared the activity of these three Ser5 proteins. Recently, Ser5 was reported to potentiate the NF- κ B signaling pathway after stimulation by TNF- α (PMID: 34520227). To test this activity, Ser5 WT or its K130R and 1-19K/R mutants were expressed with a NF- κ B-luciferase reporter in HEK293T cells and treated with TNF- α . We confirmed that TNF- α strongly activated the NF- κ B pathway, which was further enhanced by the WT protein (**Fig.S1C**). Importantly, we now show that K130R and 1-19K/R retain the same activity, indicating that this Ser5 activity does not depend on the cell surface localization of Ser5. Thus, this Ser5 activity in NF- κ B signaling is mechanistically different from its antiviral activity.

The same paper also reports that Ser5 enhances the NF- κ B signaling by interacting with TRAF6. Consistently, we now show that both K130R and 1-19K/R interact with TRAF5 as effectively as the Ser5 WT protein (**Fig.S1B**, lanes 7-9).

Collectively, these multiple pieces of evidence overwhelmingly demonstrate that our K-to-R mutants remain functionally active, indicating that they are folded properly.

Direct evidence that K130 is ubiquitinated should therefore be produced to rule out artefacts. One way to demonstrate direct ubiquitination is by identifying diGly peptides by Mass Spectrometry, which I suggest.

In last three months, we tried very hard to detect the K130 ubiquitination using this method but had no success. As shown above, we could detect the 120 AHIHNGFWFFK 130 peptide, but at a

very low frequency (1/18) by Mass Spectrometry, which explains why the K130 ubiquitination could not be detected. We speculate that once Ser5 is polyubiquitinated at K130, it is subjected to rapid turnover by the lysosomes, which makes it technically even more challenging for detection.

Nonetheless, we already presented direct and convincing evidence to demonstrate the K130 polyubiquitination. For examples, in Fig.2B, we showed that the K130R mutation selectively ablates the Ser5 ubiquitination; in Fig.2C, we showed that K130 is ubiquitinated via K33 and KK48 ubiquitin chains by using ubiquitin mutants with individual lysine mutation; in Fig.2C, we further confirmed the result from Fig.2B by using ubiquitin mutants expressing single lysine; in Fig.2D, we further demonstrated that Cul3/KLHL20 ubiquitinates this K130 residue. Collectively, these multiple pieces of evidence from complementary experiments demonstrate that K130 is indeed polyubiquitinated in cells.

2) In general, all experiments performed to test SERINC5 degradation and poly-ubiquitination are performed without a proper control protein. With the current data, it is difficult to understand whether overexpression of Ub and components of the Ub ligases complexes have a generalized and non-specific degradative effect on proteins with available lysines. The only control shown, in fact, is a mutant SERINC with no lysines, which again may be totally misfolded and in any case is intrinsically resistant to ubiquitination. A non-SERINC5 membrane protein would therefore be desirable to assess if these experiments, where proteins are ectopically overexpressed, are revealing a true physiological effect.

We have provided new evidence to confirm that the 1-19K/R mutant is properly folded and functional (**Fig.S1A, Fig.S1B, Fig.S1C**). Thus, this lysine-free Ser5 mutant is the proper control for our experiments. Importantly, our results were not only from ectopic overexpression, but also from CRISPR/Cas9 knock down (Fig.1). We also used numerous Ser5 lysine mutants and ubiquitin mutants. Altogether, we remain confident that our results reveal a true physiological effect.

Reviewer comments, second round –

Reviewer #1 (Remarks to the Author):

The authors have addressed satisfactorily most of my questions. However, I still have a minor concern. The authors show that Cul3/KLHL20 can promote K33 and K48 ubiquitination of SERINC5 with similar efficiencies and conclude that this E3 ligase promotes multibranch K33/K48 ubiquitin chains in SERINC5 at residue K130. Since this type of K33/K48 Ub branches have not been reported before, and because the authors could not confirm this hypothesis by Mass Spec, I would be more cautious when driving this conclusion. For instance, it is possible that a subset of SERINC5 molecules is K48 ubiquitinated and another subset is K33 ubiquitinated, which may be dictated by the use of different E2 enzymes by KLHL20.

Reviewer #2 (Remarks to the Author):

I appreciate the author's efforts to answer the criticism raised. However, I could not find figure S2 (mentioned in the rebuttal).
Apart from this, I am happy with the revised manuscript.

Reviewer #1 (Remarks to the Author):

The authors have addressed satisfactorily most of my questions. However, I still have a minor concern. The authors show that Cul3/KLHL20 can promote K33 and K48 ubiquitination of SERINC5 with similar efficiencies and conclude that this E3 ligase promotes multibranch K33/K48 ubiquitin chains in SERINC5 at residue K130. Since this type of K33/K48 Ub branches have not been reported before, and because the authors could not confirm this hypothesis by Mass Spec, I would be more cautious when driving this conclusion. For instance, it is possible that a subset of SERINC5 molecules is K48 ubiquitinated and another subset is K33 ubiquitinated, which may be dictated by the use of different E2 enzymes by KLHL20.

It is well known that K33 ubiquitin chains only regulate protein post-Golgi trafficking, but do not contribute to protein degradation. In addition, it is also known that SERINC5 is downregulated by Nef from the plasma membrane for endosome/lysosome degradation. Thus, SERINC5 should have both K33/K48 chains on K130 for its downregulation from the cell surface.

Nonetheless, to describe our results more accurately, we have replaced “K33/K48-branched ubiquitin chains” with “K33/K48-linked ubiquitin chains” in our abstract, results, and discussion of this revision.

Reviewer #2 (Remarks to the Author):

I appreciate the author's efforts to answer the criticism raised. However, I could not find figure S2 (mentioned in the rebuttal). Apart from this, I am happy with the revised manuscript.

We apologize for this confusion. This figure was already embedded in the previous point-by-point letter (page 2, response to Reviewer #2), allowing this reviewer to have a convenient access to read this figure. We thought to present this figure as Figure S2, but we did not.